# Factors Affecting Posterior Capsule Opacification in the Development of Intraocular Lens Materials

**DOI:** 10.3390/pharmaceutics13060860

**Published:** 2021-06-10

**Authors:** Grace Cooksley, Joseph Lacey, Marcus K. Dymond, Susan Sandeman

**Affiliations:** 1School of Pharmacy and Biomolecular Sciences, University of Brighton, Brighton BN2 4GJ, UK; m.dymond@brighton.ac.uk; 2The Ridley Innovation Centre, Rayner Intraocular Lenses Limited, Worthing BN14 8AG, UK; josephlacey@rayner.com

**Keywords:** posterior capsule opacification, pathophysiology, wound healing, lens epithelial cells, intraocular lenses, experimental models, clinical studies

## Abstract

Posterior capsule opacification (PCO) is the most common complication arising from the corrective surgery used to treat cataract patients. PCO arises when lens epithelial cells (LEC) residing in the capsular bag post-surgery undergo hyper-proliferation and transdifferentiation into myofibroblasts, migrating from the posterior capsule over the visual axis of the newly implanted intraocular lens (IOL). The developmental pathways underlying PCO are yet to be fully understood and the current literature is contradictory regarding the impact of the recognised risk factors of PCO. The aim of this review is firstly to collate the known biochemical pathways that lead to PCO development, providing an up-to-date chronological overview from surgery to established PCO formation. Secondly, the risk factors of PCO are evaluated, focussing on the impact of IOLs’ properties. Finally, the latest experimental model designs used in PCO research are discussed to demonstrate the ongoing development of clinical PCO models, the efficacy of newly developed IOL technology, and potential therapeutic interventions. This review will contribute to current PCO literature by presenting an updated overview of the known developmental pathways of PCO, an evaluation of the impact of the risk factors underlying its development, and the latest experimental models used to investigate PCO. Furthermore, the review should provide developmental routes for research into the investigation of potential therapeutic interventions and improvements in IOL design in the aid of preventing PCO for new and existing patients.

## 1. Introduction

Despite the histopathology of posterior capsule opacification (PCO) being well characterised, the molecular mechanisms underlying the pathology are still unknown [1,2,3,4]. In addition to this, current literature contains contradictions regarding the extent to which established risk factors impact PCO development. Herein, the known biomolecular pathways in PCO development are reviewed and a chronological overview of the mechanisms underlying the pathophysiology of PCO formation are presented. Moreover, this review explores the risk factors for PCO and considers their impact on PCO development. Lastly, this review examines the latest experimental models in PCO research used to investigate the next generation of medical and technological advancements for patients with PCO.

PCO is the most common complication arising from corrective surgery to treat cataracts [1,5,6,7]. In 2018, the World Health Organisation [WHO] estimated that 90 million people worldwide still live with cataract-associated blindness [8]. Each year, an additional 1–2 million people become blind, of which 75% are treatable [9]. This number is only expected to rise due to an increasingly expanding and ageing global population [9]. The Royal National Institute of Blind People (RNIB) estimated that by 2020, 695,000 people would be living with cataracts in the United Kingdom, a number that would increase by 30% between 2020 and 2030. In England alone, 330,000 cataract surgeries are performed per year [10]. Whilst in the UK cataract surgery is a routine, outpatient surgical procedure, a number of complications can occur for a subsection of patients. Approximately 20–50% of patients develop PCO and require further corrective treatment [11].

PCO presents as a secondary cataract, an agglomeration of cells over the visual axis causing a loss of acuity. During surgery, the surgeon will create an opening in the capsular bag, known as capsulorhexis, and use phacoemulsification to remove the diseased, opaque lens and lens epithelial cells (LEC) from the capsular bag before implanting an artificial intraocular lens (IOL) [12,13]. The extent of LEC removal influences the propensity towards PCO development [14,15]. The initial inflammation caused by the surgical trauma may incite the hyper-proliferation, transdifferentiation, and migration of residual LECs [16]. The transformed LECs migrate along the posterior capsule towards the anterior chamber to accumulate over the visual axis, forming a secondary cataract [1,17].

### 1.1. Pathophysiology of Posterior Capsule Opacification

The wound healing response of LECs post-cataract surgery is believed to be the first key developmental stage of PCO. Ocular inflammation is triggered as a result of the opening incision and subsequent lens cell removal [18,19]. Jiang et al. [16] showed that within the first 24 h post-surgery, the LEC transcriptome differentially expresses 19 of the 27 cataract-associated genes, some of which are markers of mesenchymal cell fate and are associated with chronic inflammatory conditions. The genes with the greatest upregulation in expression are *CXCL1, S100a9, CSF3/G-CSF, COX-2, CCL2, LCN2*, and *HMOX1* [20]. Some of these genes are involved in the production of proinflammatory chemokines such as CXCL1, alarmin S100a9, and G-CSF [16]. This response may be initiated by the surgically induced break in the blood-aqueous barrier and subsequent leakage of plasma protein into the aqueous humour (Figure 1A) [16,19]. At 48 h post operation, the elevated levels of inflammatory mediators interleukin (IL)-6, IL-1β, and IL-8 expressed by the LECs initiate and promote chronic inflammation pathways (Figure 1B) [5,21,22,23,24,25]. Migrating neutrophils and macrophages attracted by the secreted chemokines cleave the inactive precursor of the latent transforming growth factor beta (TGF-β) via proteinases in the aqueous humour [26,27]. The activated TGF-β binds to LECs via type I and type II receptor serine-threonine kinases on the cell surface [2]. This begins a cascade of SMAD proteins signalling, the messengers activated via TGFβ signalling [28,29]. These signalling pathways lead to TGF-β-gene transcription, activation of Rho GTPases, and stimulation of the PI3/Akt and MAPK pathways (Figure 1C). These events are associated with myofibroblast formation, epithelial-mesenchymal-transition (EMT)-related matrix contraction, cell differentiation, and inhibition of normal LEC pathways [2]. As a result, the LECs are stimulated to hyper-proliferate and differentiate into myofibroblasts, leading to their eventual migration over the visual axis of the IOL (Figure 1D).

From current literature, the hypothesised key mediators involved in PCO development are IL-1β, IL-8, IL-6, and TGF-β. IL-1β can be synthesised by retinal pigment epithelial cells, LECs, corneal epithelial cells, and to some extent, corneal stromal cells when triggered during inflammatory processes [25,30,31]. IL-1β promotes inflammation by acting as a proinflammatory mediator, stimulating the secretion of other cytokines such as IL-8 [24,25,32]. IL-8 can be produced by peripheral blood monocytes, endothelial cells, fibroblasts, and epithelial cells [25]. A study conducted by Ferrick [25] exemplifies the interconnectivity of IL-1β and IL-8 in augmenting the inflammatory response. Rat eye models were first injected with IL-1β. At 20 h post injection, IL-8-activated neutrophils were detected. The rat neutrophils levels then subsided after 48 h. However, this inflammatory resolution may not occur as quickly in humans. The study concluded that IL-1β produces a stronger inflammatory response with long-lasting effects due to the cascade of pathways it initiates such as activating leukocytes, instigating proinflammatory mechanisms in local cells and promoting the synthesis of cytokines such as IL-8, whereas IL-8 acts as a specific chemoattractant for neutrophils.

The presence of IL-6 during PCO development is supported by the study conducted by Nishi [23]. The authors found IL-6 expression to be significantly higher in the aqueous humour of LECs obtained during cataract surgery in comparison to medium controls. IL-6 is believed to be involved in the upregulation of extracellular matrix (ECM) synthesis, the contraction of the capsular bag, and EMT of LECs, as shown by Ma [5]. Within 24 h of IL-6 stimulation, the human LEC-B3 (HLE-B3) line initiated a significant increase in ECM synthesis and activation of the JAK/STAT3 pathway was observed [5]. Cells treated with JAK/STAT3 inhibitor WP1066 showed significantly inhibited expression of collagen, fibronectin, and TGF-β2 (*p* > 0.01). This suggests that the JAK/STAT3 pathway influences the expression of ECM proteins in HLE-B3 cells. The authors also showed that IL-6 works in synergy with TGF-β to promote EMT, since HLE-B3 cells treated with IL-6 and TGF-β2 showed significantly increased expression of the markers of EMT, alpha smooth muscle actin (α-SMA), collagen, and fibronectin in comparison to cells treated with either IL-6 or TGF-β2.

TGF-β is a well-established key mediator within PCO development [1,6,16,17,29,33,34]. The isoforms TGF-β1 and -β2 are found in ciliary processes and within the limbal epithelium where they are hypothesised to be involved in the transdifferentiation of conjunctival to corneal epithelium. As a consequence, it is believed these isoforms are locally synthesised due to their predominance in the aqueous humour [6,35]. Moreover, Nishi [36] found that LECs express TGF-β2. The binding of TGF-β to its receptor activates kinase domains within the receptor, augmenting phosphorylation cascades which initiate SMAD3 transcription factors [2,28]. The TGF-β receptor kinase has many roles within the ocular tissue, in both normal and pathological conditions [37,38]. TGF-β is important in maintaining corneal integrity, in tissue repair, and in the regulation of cell proliferation and death [37,38]. Nevertheless, hyper-activation of the growth factor can lead to an exaggerated wound-healing response and increased ECM deposition by cells, which may cause fibrotic corneal disease, fibrosis of lens and retinal epithelium, and loss of vision [6,28,35,37]. 

Short-term exposure to TGF-β can lead to long-term impact. Possible mechanisms behind this are the positive feedback loop on the upregulation of TGF-β gene expression stimulated by the exposure to TGF-β or the ability of TGF-β to bind to collagen IV, a predominant component of the capsular bag. In PCO development, TGF-β is also involved in the activation of the EMT pathway in LEC; the differentiation of the epithelial cells to migratory spindle-like myofibroblasts [29,39,40,41,42]. The activation of TGF-β inhibits the proliferation of LECs in favour of initiating transdifferentiation by upregulating the expression of α-SMA genes [6,28,43]. Therefore, α-SMA can be used as a biomarker for myofibroblasts [44]. Furthermore, TGF-β has been shown to increase the secretion of ECM components by LECs. In this process, ECM turnover is disturbed; the upregulation of fibronectin, collagen I and IV, and inhibitors of matrix metalloproteinases (MMPs) cause a greater production of ECM with reduced degradation. The increased ECM leads to capsule wrinkling and thickening of the posterior capsule [6,43,45]. The presentation of PCO can differ between individuals, depending on patient specific risk factors affecting the patients.

### 1.2. Risk Factors for Developing Posterior Capsule Opacification

PCO typically develops in the first 2–5 years post-surgery [7,11,14,47,48]. There are several risk factors that can make patients more susceptible to developing PCO. These are patient-associated risk, surgical-associated risk, and IOL-associated risk, as discussed.

#### 1.2.1. Patient-Associated Risk Factors

Studies show that patient age alters the propensity of the LECs to proliferate [7,14,47]. The younger the patient, the more LECs are within the capsular bag with greater proliferative potential [47]. Children undergoing cataract surgery can expect a 100% risk of developing PCO [49]. LECs in patients less than 40 years of age grow three times quicker than in patients less than 60 years of age [50]. Patients with diabetes have shown significant PCO development after a year follow-up. However, the severity does not differ between diabetic and non-diabetic patients over a long-term duration, suggesting that diabetes may only increase the rate of PCO development due to the initial protein-rich and inflamed tissue [14,47,51,52]. Pre-existing ocular diseases in patients such as dry-eye disease and uveitis can lead to an increased rate of PCO development and a greater likelihood of experiencing vision-threatening PCO [53].

#### 1.2.2. Surgical-Associated Risk Factors

The outcome of cataract surgery can influence the propensity towards PCO development. The removal of LECs is fundamental in preventing PCO. However, this is a difficult achievement and any number of residual LECs can mount a full PCO response [2,14,15,47]. Nevertheless, surgical interventions such as hydrodissection-enhanced cortical clean-up to remove lens substance and in-the-bag fixation to ensure IOL centralisation for optimal barrier effect can reduce PCO incidence [14,54]. An additional surgical intervention to prevent PCO development for paediatric patients is known as optic capture. Optic capture involves implanting the IOL optic through the posterior capsulorhexis opening [55,56]. This technique effectively prevents PCO as it locks the IOL in a central position preventing lateral movement and delays LEC migration due to the fusion of the anterior and posterior capsulorhexis [57]. A study conducted by Davidson [15] exemplifies how surgical technique can impact cell growth over the IOL. The surgical techniques extracapsular cataract extraction (ECCE), phacoemulsification, and phacoemulsification with capsule vacuuming were evaluated on cadaver eyes. The cell growth density over the anterior capsule was 31.6%, 16.1%, and 7.7%, respectively. When cultured in serum-free media, the phacoemulsification with capsule vacuuming group took 5.3 days longer to reach confluence. Polishing of the posterior capsule showed a reduction in PCO development as shown by Paik [58] who found that the surgeon who routinely polished the posterior capsule had a 20% PCO rate, in comparison to 30% for the surgeon who did not. Nevertheless, the impact of polishing is highly controversial and application relies on the surgeon’s preference [59,60]. Although surgical technique alone is not enough to prevent PCO, the surgeon’s skill and commitment to removal of LECs via phacoemulsification can influence severity and rate of onset.

#### 1.2.3. Intraocular Lens-Associated Risk Factors

The process of selecting the appropriate IOL is patient-specific. The choice of material, design, and function of the IOL can greatly impact risk of developing PCO. A prospective study conducted by Joshi [61] exemplifies the importance of IOL selection for patients with pre-existing conditions. A total of 1400 eyes undergoing cataract surgery with peripheral pre-existing PCO (PPPCO) were either implanted with a hydrophobic or hydrophilic IOL. PPPCO is often seen in the developing world due to the late presentation of patients and in some cases of blunt ocular trauma [61,62]. The study found PPPCO patients implanted with a hydrophilic IOL had a higher propensity towards PCO development than those implanted with a hydrophobic IOL. The author concluded that patients with PPPCO should be considered for hydrophobic IOL implantation only. Current literature offers little consensus to the impact of IOL composite material on PCO development.

##### Lens Material

IOLs have been typically constructed from polymethylmethacrylate (PMMA), silicone, or acrylic, which is further divided into hydrophobic and hydrophilic due to their differing composite monomers [63]. PMMA IOLs were the first implantable material used in the early generation of IOL technology by Ridley [64]. The biostability, low inflammatory response, compatibility with injection moulding protocols, and refractive index of 1.49 made PMMA the preferred material over glass [63,65,66]. Nevertheless, the rigidity, intolerance to elevated pressure levels, lower biocompatibility relative to silicone and acrylic, and brittleness of PMMA has led to the replacement of PMMA with more flexible materials. One driving force for this change is the surgical procedure as more flexible IOLs require smaller incision sizes, leading to a faster recovery period [19,65,66]. Silicone is a flexible hydrophobic material with a refractive index of 1.41–1.46 [63]. In the early generation of silicone IOLs, the low refractive index of silicone required thicker optics to achieve the same optical power as PMMA. However, later developments using copolymers of silicone and other monomers increased the refractive index and reduced IOL thickness [63,65]. PCO rates for silicone IOLs are reduced compared to other IOL material types due to the lower cell deposition. Nevertheless, as the capsular bag wall cannot adhere to silicone, the barrier effect of a square edge design is prevented (discussed in Lens Design) [63,66]. Another disadvantage of silicone is that the material favours bacterial adhesion increasing infection risk [63,65,67]. Alternatively, hydrophobic acrylic-based IOLs, composed of copolymers of acrylate and methacrylate, can be used. These IOLs have a high refractive index of 1.44–1.55, flexibility, and uveal biocompatibility [19,63,65]. However, these acrylic-based IOLs are easy to mark or scratch during implantation and require more attention to centre in the capsular bag when implanted [19,63]. Hydrophilic acrylic IOLs are soft, flexible, have a low tendency to scratch, and a refractive index of 1.43 [63]. Despite this, the hydrophilic composite monomers of these IOLs promote cell adhesion and some studies have suggested higher PCO rates as a result of this [63,65,67].

The extent to which IOL composite material, more specifically material wettability, impacts the onset of PCO is controversial. Recent review articles comparing hydrophobic and hydrophilic IOLs have shown some consensus on the conclusion that hydrophobic IOLs show lower PCO rates in comparison to hydrophilic IOLs [19,51,66,68,69]. Nevertheless, these clinical studies used small sample sizes and the trend of PCO development may be less disparate between hydrophobic and hydrophilic IOLs if a larger cohort size is used. Using clinical studies with smaller sample sizes may show a trend of PCO development between hydrophobic and hydrophilic which may not be valid as the impact of a single individual can greatly influence the overall conclusion. Moreover, the patients’ associated risk is not accounted for as it would be in a larger population; separating impact of the material alone would be impossible. This exemplifies an important limitation in using studies with smaller sample sizes in a meta-analysis on factors influencing PCO development. The validity of such studies can be easily questioned and can lead to misdirection of clinicians.

Examples of reviews using studies with large participant cohorts include Auffarth [69] who conducted a retrospective cohort design review to compare PCO and neodymium-doped yttrium aluminium garnet (Nd:YAG) capsulotomy rates between patients implanted with PMMA, silicone, and hydrophobic and hydrophilic acrylic IOLs. Patients were aged 50–80 years and were followed up for a minimum of 3 years. A total of 1525 patients were included (*n* = 384 PMMA; *n* = 426 silicone; *n* = 421 hydrophobic acrylic; *n* = 294 hydrophilic acrylic). The review found an overall PCO rate of 22.8% (28.3%, 21.6%, 8.9%, and 37.0%, respectively) and an overall Nd:YAG rate of 17.3% (19.3%, 16.2%, 7.1%, and 31.1%, respectively). It was concluded by the authors that hydrophobic acrylic IOLs show significantly lower PCO and Nd:YAG laser ablation rates. Another study supporting the argument that hydrophobic IOLs show lower PCO rates is a retrospective review conducted by Boureau et al. [52] who analysed patient charts to determine the requirement for Nd:YAG capsulotomy between hydrophilic IOLs (XL-Stabi, Zeiss-Ioltech, Ioltech, France) and hydrophobic IOLs (SA60AT, AcrySof, Alcon, Fort Worth, Tex or AR40E, Advanced Medical Optics Inc, Santa Ana, CA, USA). A total of 767 patients fulfilled the inclusion criteria (*n* = 263, XL-Stabi; *n* = 250, AcrySof SA60AT; *n* = 254, AR40E). The patients were aged 50–85 years and were followed up for a minimum of 36 months. The study showed a PCO rate of 52.9%, 13.6%, and 26.8%, respectively and a ND:YAG capsulotomy rate of 51.0%, 12.0%, and 25.2%, respectively. Other studies have concluded differently. Mathew [70] assessed the outcome of 3461 eyes implanted with a hydrophilic IOL (Rayner C-flex 570C). The patients were 39–93 years of age and the follow-up period ranged 5.3–29.0 months. This study found a Nd:YAG capsulotomy rate of 0.6% at 12 months and 1.7% at 24 months. Nevertheless, studies have also found there to be no difference in PCO development between hydrophobic and hydrophilic IOLs. A clinical study conducted by Bai [71] showed 60 patients implanted with a 360-degree square edge hydrophilic acrylic IOL (Rayner C-flex 570C) or a square edge hydrophobic acrylic IOL (Sensar AR40E) had no statistically significant difference in PCO grade after 24 months. There is sufficient evidence to suggest that IOL material can influence the propensity towards PCO development. However, the influence of material wettability on patient’s susceptibility to PCO cannot be fully established until additional targeted clinical studies with larger population sizes are performed.

##### Lens Design

The optic edge and haptic configuration of an IOL can influence its in-the-bag stability and the likelihood of developing PCO. The optic edge refers to the edge structure of an IOL, which is typically curved (or rounded) or square-cut (square-edge) (Figure 2A). IOLs with a square-edge design have shown a reduction in PCO development, regardless of material [2,52,71,72,73,74,75]. The square edge design is most effective when present 360° around the optic to ensure no cell growth at optic–haptic junctions [73,76]. The current theory explaining how this works is known as the barrier effect, derived from sandwich theory. Sandwich theory states that PCO development is reduced when there is maximum contact between the IOL and the posterior capsular bag wall [77,78]. The square optic edge allows the IOL to achieve the required contact with the capsular bag, preventing the migration of LECs. However, the square edge design is not effective for all patients. The design relies on contraction of the posterior capsule to meet the optic edge of the IOL establishing the barrier. This does not occur in some patients [19]. Additionally, a study conducted by Vock [79] reviewed 143 eyes over 10 years to determine Nd:YAG laser ablation rates between a round-edge silicone and a sharp-edged hydrophobic acrylic intraocular lens. The study found that 10 years post surgery, patients fitted with a sharp-edged IOL had significantly greater cumulative Nd:YAG rates than those fitted with a round-edge IOL. This unanticipated finding may be a result of late barrier failure, the delayed redivision of the capsule leaves caused by Soemmerring ring formation establishing a collagenous sealing of the capsule leaves to the optic rim. Moreover, an increase in haptic number and configuration can disturb the square edge design [80,81]. Single-piece IOLs show greater in-the-bag stability. However, they have slightly higher rates of PCO development in comparison to three-piece IOLs (Figure 2B) [81,82,83].

##### Lens Function

The function of an IOL determines the post-operative outcome in terms of visual acuity, range of vision, and likelihood of developing long-term complications. Monofocal IOLs improve visual acuity for patients. However, patients remain with one focal point and still require spectacles for near vision [65,85]. Multifocal IOLs utilise bifocal or trifocal points, allowing near, intermediate, and distance vision. However, they are consequently more likely to experience dysphotopsias and post-operative complications [63,65,85,86]. Toric IOLs are given to cataract patients with astigmatism. However, if the lenses become misaligned, the patient may experience blurred vision and require glasses [65,85,86]. In the next generation of IOL technology, the development of accommodative IOL (AIOL) will provide patients with improved visual acuity and dynamic changes in focus [65,86,87]. Nevertheless, the current commercially available AIOLs increase PCO incidence [12,88]. This is demonstrated by Sadoughi [12] who found that there was a 23% increase in PCO development for patients fitted with the AIOL Crystalens HD in comparison to patients fitted with monofocal lenses. The influence of IOL material, design, and surgical technique on the onset and severity of PCO development is too complex to differentiate each factor’s sole impact. Producing an effective treatment for all patient types is a challenging process considering the complex pathology and multitude of risk factors underlying PCO development.

### 1.3. Therapeutic Interventions for Posterior Capsule Opacification

PCO is managed with Nd:YAG laser capsulotomy. The agglomerated cells are targeted, creating an opening which restores visual acuity. A retrospective study of 806 patients found the requirement for Nd:YAG capsulotomy was 10.6% after one year, 14.8% after two, and 28.6% after four years [89]. The procedure can lead to complications such as retinal detachment, cystoid macular oedema, IOL displacement, mild anterior uveitis, and transient intraocular pressure [90]. Developing preventative measures to inhibit the biomolecular pathways of PCO could reduce the risk to patients, lower the requirement for additional treatment post cataract surgery, and ease the burden on healthcare services. Thus far, improvements in surgical technique and the square edge optic design have shown a reduction in PCO incidence [4,9,72,76,91]. Non-pharmacological methods to reduce PCO development include gene therapy and inducing osmotic changes in LECs [2]. Pharmacological approaches to preventing PCO include cytostatic drugs, anti-inflammatory drugs, and antagonists to key molecules within the developmental pathways of PCO as exemplified by Shao [92]. Shao [92] stimulated the HLE-B3 cell line SRA01/04 with TGF-β2 to induce EMT then treated the cells with fasudil, an inhibitor of the Rho-kinase activated during TGF-β signalling. Fasudil significantly reduced cell proliferation and migration, down-regulated α-SMA expression, and prevented the suppression of epithelial marker Connexin43. Nevertheless, despite extensive research into therapeutic solutions for PCO, no clinical treatment exists other than Nd:YAG.

### 1.4. Capsular Devices to Prevent Posterior Capsule Opacification Development

Capsular devices were developed to facilitate cataract surgery by improving capsular bag stability and intraocular lens centration [93,94]. The introduction of the capsular device by Hara [95] in 1991 has since led to the evolution of endocapsular devices that have aided capsular support and stability leading to a subsequent reduction in PCO development. The equator ring (E-ring) introduced by Hara [95] was a closed silicone circle with square-edge design and an inner groove to allow IOL fixation. The squared edges were hypothesised to delay PCO formation. Hara [96] compared 51 eyes, 14 of which were implanted with the E-ring, and found the E-ring significantly reduced PCO development. This design was built upon by Nishi who developed a capsular tension ring (CTR) made from PMMA with sharp rectangular edges. Although the implantation of a CTR has shown reduced PCO formation in comparison to patients implanted with an IOL alone, the discontinuous capsular bend created allows the possibility for LEC migration [93,97,98]. This CTR design was adapted into an open-capsule device for the purpose of expanding and opening the capsular bag, separating the anterior and posterior capsules [93,99]. The open-capsule device can be made of hydrophobic or hydrophilic material and is a closed ring with a square-edge design. Alon [100] showed that the open-capsule ring reduced PCO development, regardless of base material. New Zealand white rabbits were assigned either to two control groups, each implanted with hydrophobic or hydrophilic IOLs without an open-capsule ring, or to four study groups, each implanted with a hydrophobic or hydrophilic IOL and a hydrophobic or hydrophilic open-capsule ring. A clinical evaluation found a 69% reduction in the eyes implanted with the device relative to the control eyes. This reduction has been linked to the 360° squared edge and the delivery of aqueous humour to the capsule equator through windows in the rim of the device that prevents LEC migration and proliferation, respectively [93]. An additional endocapsular device was developed by Sharklet Technologies, Inc. (Aurora, CO, USA) to act as an artificial capsular bag. The protective silicone membrane has a square-edge haptic ring which provides a ridge for the IOL haptics. The signature feature of this design is the composite sharkskin-inspired microtopography which inhibits bioadhesion and has been hypothesised to prevent LEC migration [93,101]. These capsular devices, in addition to the other risk factors discussed, present the complexity of understanding and treating PCO. Experimental models are utilised in PCO research for insight into the underlying pathophysiology, testing newly developed IOL technology in aid of device optimisation, and investigating potential therapeutic interventions.

## 2. Experimental Models to Investigate Posterior Capsule Opacification

### 2.1. In Vitro Models

Whilst in vitro models are not representative of the physiological environment they have been used as a rapid and cost-effective route to ascertain the molecular mechanisms underlying PCO formation. Additionally, cell models are typically used in the first stage of investigating possible therapeutic interventions due to the reduced ethical constraints and ability to provide fundamental toxicology data prior to using relatively more expensive in vivo and ex vivo models. Cell culture models have shown the role of fibronectin, growth factors such as TGF-β, and enzymes including aldose reductase in the development of PCO [102,103,104]. The creation of three-dimensional cell culture models has broadened the application of in vitro models to encompass structural replicability.

#### 2.1.1. Two-Dimensional In Vitro Models

Cell cultures are used as in vitro models as the cultures provide a dynamic growing system [17]. Cultures are comprised of primary or immortalised cell lines. Chick primary cell cultures derived from chick lenses have been used to model the lens fibre cell differentiation that occurs during the development of Soemmerring’s ring in PCO and the early stages of growth factor signalling. This is demonstrated by VanSlyke [102] who used a chick primary cell line to examine the role of fibronectin within the transformation of lens cells post-surgery. Chick embryonic lens cells from primary cell cultures were either plated on laminin-control or fibronectin-coated wells. Western blot analysis was conducted to quantify the expression of SMAD isoforms, markers of TGF-β signalling. The study showed the cells incubated with the fibronectin coating expressed pSMAD3 at 1.9 ± 0.23-fold increase (mean ± SEM; *n* = 5; *p* = 0.001) in comparison to the laminin-control at day 3. This study suggests the potential involvement of fibronectin in TGF-β signalling.

In addition to primary cell lines, the human lens epithelial cell line HLEB3 is one of the most established cell lines for research into the development and treatment of PCO [1,105,106,107]. HLE-B3 cells possess receptors for growth factors and when incubated with these factors, elicit fibrotic pathways shown during PCO development. Wertheimer [103] used the HLE-B3 cell line to investigate the use of erlotinib, an inhibitor of epithelial growth factor (EGF) signalling. The study used tetrazolium dye-reduction assay (MTT) and Live-Dead assay to determine the toxicity of different concentrations of erlotinib on the cells. Erlotinib became toxic to the lens cell line at 100 μM. Chemotactic migration assessed by Boyden chamber assay and chemokinetic migration assessed by time lapse microscopy found that Erlotinib significantly reduced cell migration (*p* = 0.004 and *p* = 0.001, respectively). Wertheimer [108] continued this work by submerging IOLs within solutions of erlotinib and found that after 24 h exposure, complete cell coverage took 5.9 to 8 days longer than the control IOL.

#### 2.1.2. Three-Dimensional In Vitro Models 

The development of three-dimensional in vitro models of PCO provides greater functional and structural applicability as shown by Plüss [106] who created a three-dimensional in vitro model of HLE-B3 cells by seeding the cells within 96-well plates and centrifuging to aggregate the cells. The spheroid structure was established by day 1 and was maintained up to day 26 (Figure 3). The expression of key epithelial markers N-cadherin (CDH2) and αβ-crystallin (CRYAB), as defined by the study, were confirmed by reverse transcription-quantitative polymerase chain reaction (rt-PCR). The study confirmed the expression of these markers was stable over a period of 15 days. Their gene expression analysis showed the repression of Ephrin type-A receptor (EphA2), involved in maintaining lens transparency, and the significant upregulation of ACTA2, a marker of EMT during PCO development.

In summary, cell cultures as an in vitro model provide insight into the molecular mechanisms of PCO whilst being easily accessible for all laboratories due to their lack of reliance on the limited and often expensive supply of donor tissue. However, despite the development of 3D cell models, there is still restricted applicability of isolated cell cultures to in vivo and ex vivo models. The in vivo models used in PCO research administer treatment or surgical interventions on animal donor eyes and examine the impact on the donor pre- and post-dissection. Unlike in vitro models, in vivo models allow real-time inflammatory responses and provide insight into the impact of surrounding tissue.

### 2.2. In Vivo Models

In vivo models can be exploited in many aspects of PCO research. Such applications include the investigation of intraocular lenses, underlying biomolecular mechanisms, and the efficacy of surgical interventions, and the testing of therapeutic inhibitors of developmental pathways [17,109,110,111,112]. Animal donors include murine, rabbit, and porcine species. However, caution is required when comparing and extrapolating pathological responses between animals and humans due to differences in species biology [1,17]. Furthermore, each animal donor type has its own limitations.

#### 2.2.1. Murine

Murine models include rat and mice donors. These models cannot be utilised to evaluate IOL technology due to size restriction. Nevertheless, mice models include transgenic variations. Knock-out mice show the involvement of key molecules in the pathophysiology of PCO which can be built upon to develop inhibitory treatments [113]. Kubo [114] exploited murine models to show the role of tropomyosin in PCO development. Extracapsular lens extraction was performed on mice and rat eyes and the expression of tropomyosin (Tpm) isoforms was examined. The study used SDS-PAGE and Western blot analysis to quantify Tpm1 and Tpm2 expression. For the eyes with cataracts, the bands were stronger and immunohistochemistry showed that Tmp1/2 was found predominantly in the cytoplasm and surface lens fibre. The study continued by relating a higher level of Tmp isoforms to the induction of fibroblastic changes and up-regulation of EMT marker, α-SMA. The study concluded that Tpm1α/2β could be used as markers for EMT in LECs. In addition to providing insight into unknown molecular mechanisms, murine models can be used to investigate inhibitory therapeutic interventions as demonstrated by Lois [41] who used rat models to investigate the role of TGF-β2 and anti-TGF-β2 antibody. The rat eyes were injected with either solutions of 1 ng/mL TGF-β2, 1 mg/mL human monoclonal TGF-β2 antibody, or 5.2 mg/mL control IgG4 antibody. Post-operative clinical evaluation and immunohistochemistry analysis showed no significant difference in PCO development or α-SMA staining between the conditions. The authors suggested this to be due to the other isoforms of TGF-β compensating for the blocked TGF-β2 activity.

#### 2.2.2. Rabbit

Rabbit models can be used to examine the pathophysiology of PCO as demonstrated by the study conducted by Gerhart [111] who investigated the prevalence of Myo/Nog cells, involved in normal morphogenesis of the lens and retina, within the ciliary process and posterior capsule following cataract surgery. Cataract surgery was performed on the rabbit eyes and the PCO grade was evaluated at day 30. At 24 h post surgery, the number of Myo-Nog cells had significantly increased in the equatorial region of the lens [*p* = 0.0001], the ciliary processes (*p* = 0.0002), and some of the zonule fibres. At day 30, a correlation between the elevated Myo/Nog cell density and the average PCO grade of 2.1 ± 0.8 (*n* = 13 eyes) was found. The Myo/Nog cells overlying the capsule wrinkling expressed α-SMA, a biomarker of myofibroblastic cells involved in PCO [44]. The study concluded that Myo/Nog cells contributed to PCO development for some adults and children and could be targeted to prevent capsule wrinkling and PCO. 

Rabbit models can also be used to investigate potential treatments and unlike murine models, can be used to test new IOL technology [17]. This is exemplified by Han [109] who used a rabbit model to evaluate the treatment of an antiproliferative-drug-eluting IOL. IOLs were coated with doxorubicin-loaded chitosan nanoparticles layered with heparin, to decrease cell proliferation and adhesion, respectively. The IOLs were implanted into eight-week-old male rabbits and euthanised by air embolism at 10 weeks. The severity of PCO development was assessed from grade 0 to 3 via slit lamp micrographs. The findings showed that the coating was able to reduce Soemmering’s ring formation and cell proliferation. Despite the insight that rabbit models have provided, many studies have shown the different responses between rabbit and human models [17,115,116]. Rabbit immune responses are quicker and have different key pathways; therefore, these animals may respond to therapeutic interventions that would not reflect in humans [17,116]. As such, testing newly developed IOL technology and treatments in rabbit models requires the confirmation that the molecules under investigation are present in humans and rabbits.

#### 2.2.3. Porcine

Porcine models present similar anatomical, physiological, and functional features to those found in humans, providing a niche intermediate between mice and humans [117,118,119]. The porcine genome is three times closer to the human in comparison to the mouse [119]. Porcine models have been used to elicit underlying biomolecular mechanisms as shown by Ma [113] who used a porcine capsular bag model to determine the role of gremlin within PCO development to establish its potential role as a preventative treatment. Gremlin has been shown to play a role in some fibrotic diseases. However, the mechanisms underlying gremlin’s ability to induce EMT and ECM production by LEC are still unknown [113]. In the study, ten pig eyes were treated with either 200 μg/L gremlin or control media for 14 days and cell proliferation was observed via inverted microscope and immunocytochemistry. The study showed that gremlin induces expression of α-SMA, promoting the EMT pathway. Porcine models can be used to further investigate newly developed IOL technology and preventative therapeutics prior to human models. Nevertheless, the use of pig eye donors are not as established as murine, canine, and primate models and have yet to be utilised fully within PCO research [117,118,119]. Despite the crossover of many of the applications of each animal donor type, there are important distinctions between the types. Murine, rabbit, and porcine models can all provide insight into the key molecules contributing to PCO development and be used to test therapeutic interventions. However, rabbit and porcine models alone allow IOL implantation. In addition, pig tissue is the closest structural and functional match to human tissue than any of the other animal donors.

### 2.3. Ex Vivo Models

Ex vivo models have a benefit over in vivo models as the explant system maintains the natural structure of the surrounding tissue; therefore, these models provide insight into the localised impact of treatments with less stringent ethical constraints [17,120]. A study conducted by Kassumeh [121] exemplifies this as this study investigated slow-releasing methotrexate (MTX)-loaded poly (lactic-co-glycolic) biomatrices as a IOL coating to reduce PCO. MTX was selected as the best candidate from a review of potential pharmacological drugs that were cross-referenced against those approved by the FDA or EMA. The study found no toxic effects on the corneal cell line CEC-SV40. Moreover, the study performed open-sky cataract surgery on twelve cadaver eyes then implanted the coated IOLs. The capsular bags were kept pinned in culture flasks and kept under standard cell culture conditions. At 9.3 days, the control IOLs showed full cell coverage whereas at this time point, the MTX-coated IOLs showed residual LECs were only visible at the outer edges and took 51.0–51.3 days to achieve full coverage. The application of ex vivo models is further supported by D’Antin [122] who used human eye donor tissue to investigate the treatment of hydrogen peroxide and distilled water to prevent cell proliferation within PCO development. The capsule bag-ciliary body complex was dissected from the donor eyes and transferred to sterile petri dishes onto a silicone ring mould where the iris was removed. Continuous circular capsulorrhexis, hydrodissection, and hydroexpression was performed before individual treatment using a silicone irrigation ring (Figure 4). The study found that both treatments significantly delayed cell growth by day 28 whereas the control donor eyes reached cell confluence by day 11.

In contrast, Taiyab [29] used rat ex vivo models to investigate the role of TGF-β in the EMT pathway within PCO development. In their first study, the explant models were treated with TGF-β with or without SIS3, a SMAD3 inhibitor. The untreated samples showed no α-SMA and stress F-actin fibre expression whereas lens explants treated with TGF-β displayed a greater staining of these markers. The explants treated with TGF-β and SIS3 reduced the expression of α-SMA and stress F-actin fibre expression which corresponded with the positive control ICG-001, an inhibitor of TGF-β-induced EMT. Following on from this work, the study investigated the E-cadherin/β-cadherin complex within the cell periphery that undergoes rearrangement during EMT-related PCO. On treating the explants with TGF-β, a reduction in E-cadherin and increased nuclear translocation of β-catenin was observed which led to a loss in cell structure. With the addition of SIS3, the staining of E-cadherin did not alter in comparison to the untreated control explants and only a partial reduction in β-catenin nuclear translocation was seen.

Ex vivo models have also been used to improve the reproduction of the clinical environment of PCO as shown by Eldred [123] who used a graded culture media in a human capsular bag model to assess the impact of IOL design on its propensity to initiate PCO development. Mock surgeries were completed on donor human eye tissue after which the eyes were implanted with either Hoya Vivinex or Alcon Acrysof lenses. The capsular bag and surrounding ciliary tissue were secured to a silicone ring within a culture disk. The capsular bag was kept in a graded culture system to reflect the clinical environment, beginning with 5% human serum and 10 ng/mL TGF-β2 and reducing to serum-free media by day 15–28. A matched pair design was used to assess the implanted IOL and graded culture serum. The findings showed that the graded culture produced quicker posterior coverage, matrix contraction, and significantly increased light scatter. Cell growth over the central optic was greater in the Alcon Acrysof and significantly increased light scatter occurred.

Experimental models provide inexpensive, readily available replicas of the ocular tissue environment to reproduce the pathology of PCO and investigate therapeutic interventions and new IOL technology. Nevertheless, the differences in the wound-healing response and inflammation pathways between species prevents direct comparison to human studies. Experimental models can be exploited prior to clinical studies and present alternative, easily accessible methods for scientific investigation; however, clinical studies utilise a longitudinal design to provide important insight into the dissemination of PCO and the long-term risk factors of novel medical devices and treatments.

### 2.4. Clinical Studies

Building upon the evidence of preliminary experimental models, clinical studies provide a final evaluation of newly developed IOL technology or therapeutic interventions by using human participants and longitudinal reviews [124]. Clinical studies have been used in PCO research not only for the previously discussed applications but also in determining distribution of PCO and Nd:YAG capsulotomy rates. Nevertheless, these studies also present their own limitations and their specific scope or population demographic can be difficult to translate to the general population, preventing broad applicability. The different study designs and outcomes of recent clinical studies are discussed hereafter.

#### 2.4.1. Surgical Technique and Nd:YAG Laser Ablation Rates

Clinical studies can be important in determining the efficacy of a new surgical technique as shown by Kelkar [125] who evaluated the precision pulse capsulotomy (PPC) technique during phacoemulsification. The study included 99 patients, 123 eyes in the final analysis. This study did not state the nationality of the patients and a high percentage of the patients were in one age group, of a disproportionally older age range. Moreover, the use of two surgeons presented variability in the performance of the surgical technique under investigation, which is not discussed further. Nevertheless, the study reports on all complications that arose during surgery. The study also discusses previously reported analysis of this technique and provides an in-depth comparison between their findings and previously published data. Although the conclusion on the use of PPC in achieving a “perfectly round capsulorrhexis” is encouraging, the small population size limits this finding, as acknowledged by the authors.

Clinical trials can also provide insight into the epidemiology of Nd:YAG laser capsulotomy rates. This insight provided by clinical trials is an important tool in determining populations most at risk of developing PCO. However, clinical studies that have a median follow-up of less than 2 years do not provide a sufficient timeframe for PCO development [47]. Therefore, these studies would not accurately determine Nd:YAG laser ablation rates as sufficient time has not passed for PCO to develop in the participants. This is exemplified by Ursell [11] who presented a retrospective study to determine three-year incidence of Nd:YAG capsulotomy in relation to the IOL biomaterial; AcroSof hydrophobic (SN60AT/WF, MA30/60AC, MA50/60BM), non-AcroSof hydrophobic (TECNIS ZCB00/ZA9003) or non-AcroSof hydrophilic (AKREOS Mics MI60/Adapt, SOFTEC HD/1, C-FLEX 970C, SUPERFLEX 920H, INCISE, 570H Rayner). Data were collected using Medisoft electronic medical records. The study presented the baseline characteristics of each study group, which was accounted for in the logistic regression analysis. Their results showed the overall Nd:YAG laser ablation incidence rate over the three years was 6.4%. The incidence was two times lower for eyes implanted with AcrySof (2.2–2.7%, *n* = 322) relative to the Non-AcrySof hydrophobic (4.1–4.7%, *n* = 843) and four times lower for eyes implanted with Non-AcrySof hydrophilic (10.5–11.3%, *n* = 2157). The study’s experimental design was limited as patients who were ≤65 years were excluded, preventing any generalisation to younger cataract patients. However, the study had a three-year follow up period, establishing sufficient time for PCO to develop in the eye dataset, and the large cohort size of over 50,000 eyes presented a representative population for generalisation of the study’s findings.

#### 2.4.2. Evaluation of Intraocular Lenses

In addition to providing insight into the demographics affected by PCO, clinical studies have additional applications in evaluating newly developed IOL technology when released to a large population. Nevertheless, with the different brands and suppliers of IOL technology, finding a large cohort size is difficult as shown by Van [126] who completed an observational study of 143 patients, 226 eyes at final analysis to investigate the requirement for Nd:YAG for patients implanted with an enVista^®^MX60 IOL. The study did present appropriate exclusion criteria including the exclusion of patients with <24-month follow-up. However, the participant cohort was not representative due to the disproportionate representation of females (29.4/70.6% M/F) and mean age of 80.7 ± 8.3 years. Their primary outcome measures included requirement for Nd:YAG laser ablation and formation rate of glistenings; microvacuoles filled with aqueous humour trapped in the IOL. They found a 3-year cumulative incidence of Nd:YAG laser ablation of 2.2% and no reports of glistenings development. However due to their participant cohort, the results lack validity to represent the larger population.

In comparison to Van [126], Zhao [68] conducted a metanalysis from a variety of databases to determine a correlation between the wettability of an IOL material and the propensity towards PCO development. Metanalysis requires a variety of sources to collate findings; therefore, overcoming the challenge of a small participant size. The analysis investigated PCO rates between hydrophobic and hydrophilic acrylic IOLs. The metanalysis included only age-related cataracts and excluded any case reports and personal communications. The authors used the Cochrane Collaboration tool to report any bias of the studies included. The review selected 55 eligible studies out of a possible 503 studies, which presented a final total of 889 eyes/patients. Of this group, a broad age range and follow-up period (1–9 years) was included. Their findings suggested hydrophobic IOLs may have lower Nd:YAG rates and PCO score. Nevertheless, the metanalysis included studies with small participant sizes and studies with different IOL designs, which has been shown to impact PCO development. The authors acknowledged these limitations and suggested further analysis should use long-term randomized controlled studies with large sample sizes to ensure the study design does not influence the findings reported.

#### 2.4.3. Investigating Potential Therapeutic Interventions

Clinical studies can be further used to determine the effectiveness of treatments although there are a limited number of effective therapeutics to date. Hecht [127] presented a retrospective cohort study of 13,368 patients to examine the impact of steroids and NSAID therapy on PCO rates. The mean follow-up was 22.8 ± 15.7 months; a sufficient time period for PCO to develop. The results were confirmed against a second independent dataset. The participants’ variables were accounted for and were included within the dataset presented. However, a large number of surgeons were used and their referral of steroid treatment or NSAID therapy introduced subjectivity into the findings. Nevertheless, the study presented a detailed discussion and acknowledged limitations of the retrospective design of the study. The conclusion that post-treatment of steroids alone can significantly decrease PCO development, with no difference found for the additional use of NSAID therapy, could be generalised to a larger population due to the large cohort and time period of investigation within this study. Alongside the restricted availability of treatments, clinical studies are limited to small population cohorts due to the stringent exclusion criteria used specifically in human clinical trials when investigating new treatments. This is exemplified by the study conducted by Rabsilber [128] who investigated the long-term outcomes of the treatment of sealed capsule irrigation with distilled water. Only 17 patients were not excluded, and the follow-up period was 24 months. The treatment was administered to one eye and the other was used as a control. The study found the treatment did not significantly prevent PCO development. However, as in any study with a small population size, further work is required to support the conclusion.

In the last 20 years, PCO research has facilitated a decrease in patients affected by PCO as a result of the introduction of a square-edge design and improvements in surgical techniques. The experimental models discussed in this review have provided insight into the complex developmental pathways of PCO and advancements in IOL technology, allowing long-term restoration of visual acuity and lens function post surgery. A summary of the experimental model design types with their current applications to investigate PCO is shown in Table 1. In future studies, care should be taken to ensure appropriately powered studies and follow-up periods. A minimum of ≥2 years is required to ensure sufficient time has passed for PCO presentation to have begun. Moreover, there is a need for studies with a ≥10 year follow-up to show unexpected long-term complications such as the late barrier failure, an urgent requirement considering that these complications are more likely to occur as life expectancy rises and cataract surgery is performed earlier in patients [79]. The use of these study design requirements will avoid the dissemination of invalid conclusions, leading to contradictions of known PCO pathways and risk factors. These challenges can be overcome by using retrospective or metanalysis designs which collate a larger population cohort from multiple studies and provide conclusions that can be used by clinicians to improve the treatment of cataracts and IOL selection in the aid of preventing PCO development for new and existing patients.

## 3. Conclusions

In conclusion, this review presented a revised overview of the developmental pathways of PCO, from the initial wound-healing response of the LEC to the formation of a secondary cataract. Additionally, the review examined the individual impact of the risk factors underlying PCO development, establishing the square optic edge as an effective design to prevent PCO and provided clarity on the current understanding on the impact of the IOL material’s wettability. Finally, this review demonstrated the latest experimental models used in PCO research and how the model types can be exploited for different applications. Future work should focus on developing current understanding of the pathophysiology of PCO, providing in-depth analysis of the risk factors leading to PCO, specifically IOL design and function and conducting clinical studies with larger population sizes to ensure reviews provide valid and appropriate conclusions.

## Figures and Tables

**Figure 1 pharmaceutics-13-00860-f001:**
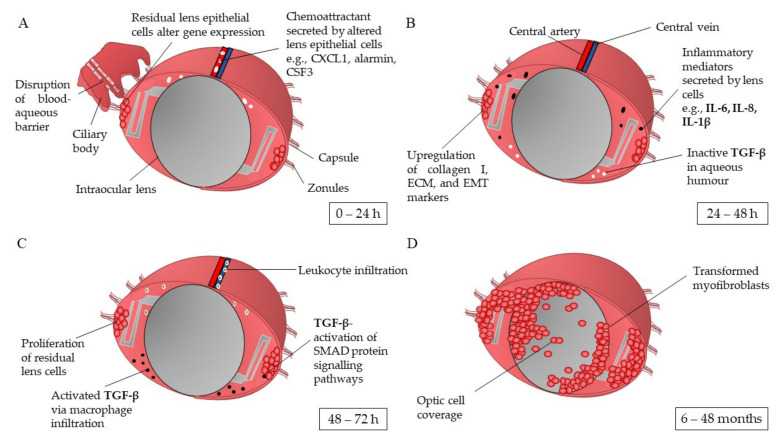
Schematic overview of the capsular bag post cataract surgery in the development of posterior capsule opacification. (**A**) Stimulated by surgery trauma, residual lens epithelial cells undergo the wound healing response, alteration of gene transcriptome and expression of chemoattracts targeting inflammatory mediators and innate immune cells. (**B**) Upregulation of inflammatory mediators, i.e., IL-1β expressed by lens cells undergoes autocrine signalling to initiate the synthesis of collagen I, extracellular matrix (ECM), and epithelial-mesenchymal transition (EMT), proteins and markers. (**C**) Residual lens epithelial cells start to proliferate; leukocyte infiltration attracted by the high levels of chemo attractants and inflammatory mediators activate dormant transforming growth factor beta (TGF-β) residing in the aqueous humour. TGF-β activates SMAD3 signalling pathways in the lens cells, stimulating PI3/Akt, Rho GTPases, and MAPK pathways. (**D**) The transdifferentiated lens cells migrate over the intraocular lens’ optic. IL: interleukin; TGF-β: transforming growth factor beta [2,5,6,16,17,20,21,25,26,37,46].

**Figure 2 pharmaceutics-13-00860-f002:**
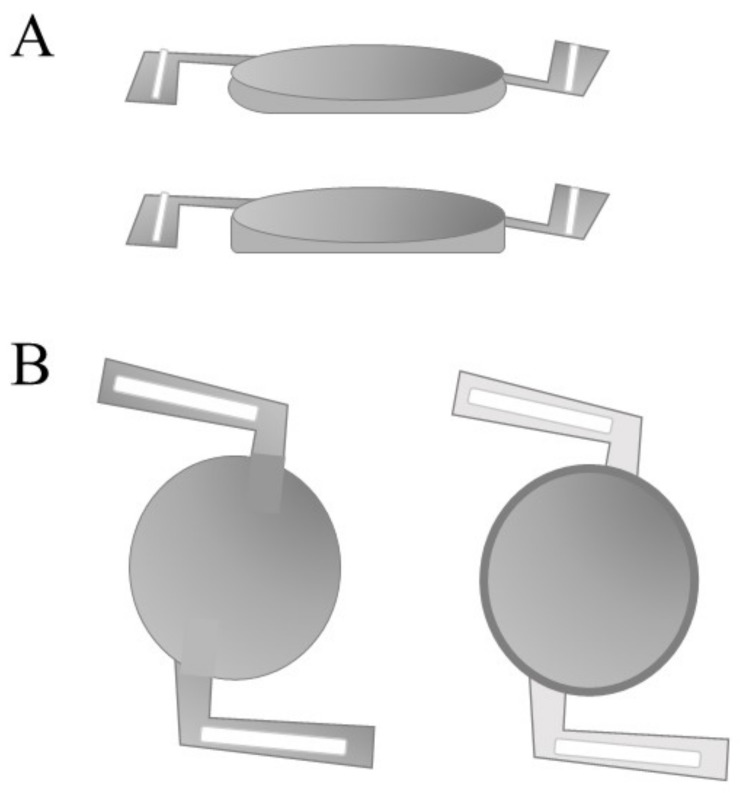
(**A**) Schematic design of an intraocular lens with a rounded optic edge (top) and squared optic (bottom). (**B**) Schematic design of the single-piece IOL (left) and three-piece IOL (right) [75,84].

**Figure 3 pharmaceutics-13-00860-f003:**
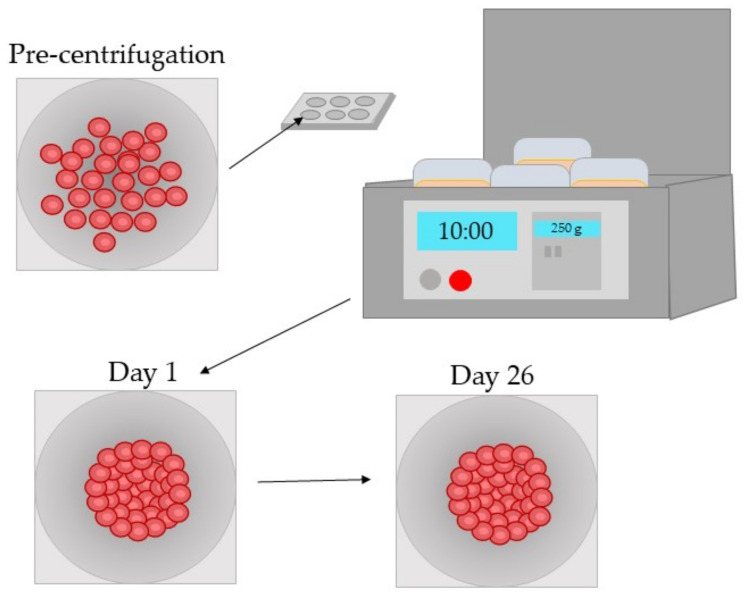
Three-dimensional in vitro spheroids formed by human lens epithelial cell line. Cells were centrifuged at 250× *g* for 10 min immediately post-seeding in a microplate. The spheroids were maintained for 26 days. Stable expression of key markers N-cadherin (Hs00983056_mL) and αβ-crystallin (Hs00157107_mL) was seen up to day 26 as determined by reverse transcription-quantitative polymerase chain reaction [106].

**Figure 4 pharmaceutics-13-00860-f004:**
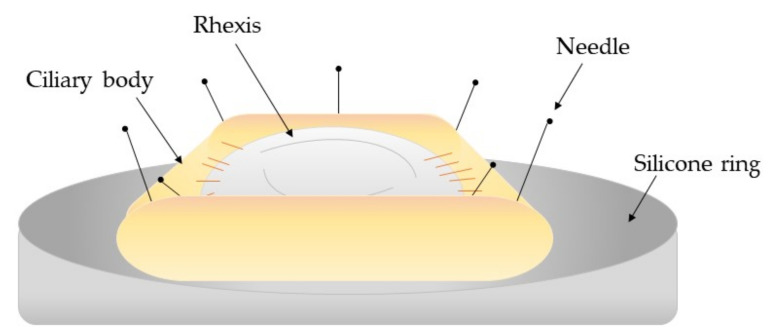
Schematic diagram of the capsule-ciliary body complex attached to a silicone ring mount [122].

**Table 1 pharmaceutics-13-00860-t001:** Overview of the applications of the experimental models of posterior capsule opacification. TGF-β: transforming growth factor beta; EMT: epithelial-mesenchymal transition; LEC: lens epithelial cells.

Model Type	Tissue Donors	Application	References
In vitro	Chick, rat, human, mouse	Three-dimensional modelling	[106,107,123,129]
		Role of cytokines and inflammation	[5,16,102]
		Investigation of intraocular lenses	[105,108,109,130]
		Inhibition of molecular pathways	[92,103,121,131,132]
		Surgical technique evaluation	[15]
In vivo	Porcine, murine, human, rabbit	Investigation of EMT pathway	[114,133]
		Investigation of intraocular lenses	[109]
		Investigation of molecular pathways	[110,111]
		Inhibition of molecular pathways	[41,112,132,134]
		Role of cytokines	[25]
		Surgical technique evaluation	[99,100]
Ex vivo	Human, rat, porcine, canine, chick	Inhibition of molecular pathways	[112,121,132,134,135]
		Investigation of molecular pathways	[29,136]
		Investigation of intraocular lenses	[108,137,138]
Clinical studies	*n*/a	Identifying risk factors	[47,89]
		Investigation of therapeutic interventions	[127]
		Investigation of intraocular lenses	[11,12,53,68,69,70,121]
		Surgical technique evaluation	[58,125]

## Data Availability

A full list of references is compiled and attached to this manuscript.

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
