# Peer review of "Factors Affecting Posterior Capsule Opacification in the Development of Intraocular Lens Materials"

_pharmaceutics, 2021, doi:10.3390/pharmaceutics13060860_

Round 1

Reviewer 1 Report

The review addresses an important and complex topic, which is the development of posterior capsule opacification after cataract surgery. The authors discussed relevant factors related to PCO development and reviewed appropriate literature. The pathology of PCO was well addressed and a wide variety of factors affecting PCO was covered. It’s a needed and interesting review that addresses a topic not found in recent literature. But there are still some details that need to be improved. A few comments are listed below: 

  • There are several grammatical and punctuation mistakes, as well as many sentences that are rather too long and confusing. Please, proofread the manuscript thoroughly to correct the mistakes. Change the sentences to improve readability and clarity. Here are some examples:

Line 50-51 - During surgery, the surgeon will create an opening the capsular bag, known as capsulorhexis and use phacoemulsification (..)

Line 99: whereas IL-8 acts as a specific chemoattract for neutrophils.

Line 168-170: Pre-existing ocular diseases in patients such as dry-eye disease and uveitis, leads to an increased rate of PCO development and greater likelihood of experiencing 169 vision-threatening PCO (54).

Line 214-217: Nevertheless, the rigidity, intolerance to elevated pressure levels, lower biocompatibility relative to silicone and acrylic and brittleness of PMMA has led to the replacement of PMMA with more flexible materials.

Line 259-261: Another study supporting the argument that hydrophobic IOL show lower PCO rates is a retrospective review conducted by Boureau et al. (53)

Line 164-167: sentence is rather too long, might consider splitting into two.

Line 188 – 190: add comma to separate the sentences.

Line 202-204: sentence is too long

Line 345-346: Nevertheless, despite extensive research into therapeutic solu-345 tions for PCO, there is yet to be any viable cure beside Nd:YAG.

Line 348-349: Capsular devices was developed to facilitate cataract surgery by improving capsular 348 bag stability and intraocular lens centration (93,94).

  • Line 190-191: The authors did not add any reference indicating the controversy related to polishing the posterior capsule. References should be added and briefly discussed.

  • Page 4 – figure 1 caption: The figure shows events occurring within the first 48h only in A and B, but C and D show processes happening until 48 months post-surgery. Please rewrite the first sentence of the figure caption to include the period described in C and D.

  • Line 205 – 207: References showing this controversy should be cited when stating that “the impact of the IOL composite material is controversial”.

  • Lines 240, 242, 247: I suggest using “small sample sizes” instead of “small participant sizes”.

  • Line 273-276: Since the authors are discussing clinical studies with large samples, the sample size of Bai et al (72) should also be added to the text.

  • Line 286-287: Please correct the symbol for 360 and review the sentence grammar.

  • Line 269: The authors say that Mathew et al (70) and Bai et al (72) studies concluded the opposite of the references 69 and 53, which showed lower PCO and Nd:YAG rates for hydrophobic IOLs compared to hydrophilic ones. However, hydrophobic IOLs were not evaluated in Mathew et al work, and, Bai et al found no significant difference in PCO for both types of IOL. These conclusions are different, but not the opposite of the other studies. Also, data from the reference 71 (Werner et al) was used to compare the percentage reported in Mathew et al. Werner study was conducted in rabbits, and with a small sample size when compared to the other studies, which might be important to mention. Thus, please consider using other references and/or changing the wording in order to be consistent with the results presented for hydrophilic IOLs.    

  • Page 7 – figure 2B caption: “Left” was mentioned twice and “right” was not indicated.

  • Line 403-414: This paragraph discusses some relevant information regarding the in vitro models. However, the ideas are not well organized and clear. Please, rewrite the paragraph to make it more readable and improve clarity.

  • Line 432, figure caption. Is this magnification related to the actual dimension of the cells?

  • Line 436-437: Could the authors please explain better which natural physiological structures you are referring to? Also, the reviewer suggests explaining this restrict applicability of isolated cell cultures to these natural physiological structures?

  • Line 466-472: The conclusion reported in Lois et al work is not clear in the manuscript. Please, explain it and give a brief discussion regarding α-SMA staining not noted by the authors.

  • Please provide the reference for the statement in line 496: “Rabbits’ immune responses respond quicker and with different key pathways”.

  • Line 529-537: The methodology employed in Kassumeh (119) is not clear in the manuscript. Please rewrite the paragraph to improve its structure and clarity.     

  • Line 542-543: The reviewer suggests describing how the treatments were given to evaluate hydrogen peroxide and distilled water to prevent cell proliferation within PCO development.

  • 621-623. When presenting the results, the authors used a different name for the IOLs from the ones used in the methodology. Please consider using the same group names.

Reviewer 2 Report

In this review an  updated overview of the  developmental pathways of PCO, and on the connected risk factors is given.

Also the different in vitro, in vivo and ex vivo models are exhaustively described. 

I have no particular suggestion to give to the Authors who were able to organize literature data in a well-constructed paper which provides insight into our  current knowledge of PCO and discuss putative management of PCO from IOL design to pharmacological interventions undelining also the need to plane  clinical studies with larger population sizes to ensure  appropriate conclusions. 

Reviewer 3 Report

While this is a review article and as such devoid of specific recommendations, some comments concerning the current thrust of the various investigative modalities and the direction of current research would be welcome.

Since it is a review it compiles scientific information from various sources which is appropriate. Yes origionsl since it is the only comprehensive source to do so yes the scientific information provided is accurate

Round 2

Reviewer 1 Report

The authors addressed all comments made in the first round of the review process and the necessary changes to the manuscript were made. I believe that the manuscript can be accepted in its current form.